# Fetal Brain-Derived Exosomal miRNAs from Maternal Blood: Potential Diagnostic Biomarkers for Fetal Alcohol Spectrum Disorders (FASDs)

**DOI:** 10.3390/ijms25115826

**Published:** 2024-05-27

**Authors:** Nune Darbinian, Monica Hampe, Diana Martirosyan, Ahsun Bajwa, Armine Darbinyan, Nana Merabova, Gabriel Tatevosian, Laura Goetzl, Shohreh Amini, Michael E. Selzer

**Affiliations:** 1Center for Neural Repair and Rehabilitation (Shriners Hospitals Pediatric Research Center), Lewis Katz School of Medicine, Temple University, Philadelphia, PA 19140, USA; monicahampe@gmail.com (M.H.); dgm438@gmail.com (D.M.); ahsunbajwa@gmail.com (A.B.); nmerabova@gmail.com (N.M.); dr.tatevosian@gmail.com (G.T.); 2Department of Pathology, Yale University School of Medicine, New Haven, CT 06520, USA; armine.darbinyan@yale.edu; 3Medical College of Wisconsin-Prevea Health, Green Bay, WI 54304, USA; 4Department of Obstetrics & Gynecology, University of Texas, Houston, TX 77030, USA; laura.goetzl@uth.tmc.edu; 5Department of Biology, College of Science and Technology, Temple University, Philadelphia, PA 19122, USA; ashohreh@temple.edu; 6Department of Neurology, Lewis Katz School of Medicine at Temple University, Philadelphia, PA 19140, USA

**Keywords:** brain development, fetal alcohol syndrome, miR-9, fetal eye, exosomes

## Abstract

Fetal alcohol spectrum disorders (FASDs) are leading causes of neurodevelopmental disability but cannot be diagnosed early in utero. Because several microRNAs (miRNAs) are implicated in other neurological and neurodevelopmental disorders, the effects of EtOH exposure on the expression of these miRNAs and their target genes and pathways were assessed. In women who drank alcohol (EtOH) during pregnancy and non-drinking controls, matched individually for fetal sex and gestational age, the levels of miRNAs in fetal brain-derived exosomes (FB-Es) isolated from the mothers’ serum correlated well with the contents of the corresponding fetal brain tissues obtained after voluntary pregnancy termination. In six EtOH-exposed cases and six matched controls, the levels of fetal brain and maternal serum miRNAs were quantified on the array by qRT-PCR. In FB-Es from 10 EtOH-exposed cases and 10 controls, selected miRNAs were quantified by ddPCR. Protein levels were quantified by ELISA. There were significant EtOH-associated reductions in the expression of several miRNAs, including miR-9 and its downstream neuronal targets BDNF, REST, Synapsin, and Sonic hedgehog. In 20 paired cases, reductions in FB-E miR-9 levels correlated strongly with reductions in fetal eye diameter, a prominent feature of FASDs. Thus, FB-E miR-9 levels might serve as a biomarker to predict FASDs in at-risk fetuses.

## 1. Introduction

Prenatal exposure to ethanol (EtOH) causes variable abnormalities known as fetal alcohol spectrum disorders (FASDs), of which the most severe and multisystemic is fetal alcohol syndrome (FAS), which is the leading cause of the only known preventable neurodevelopmental impairment. The estimated global prevalence of FASDs among the general population is 7.7 cases/1000 individuals. In fact, FASDs’ prevalence is apparently highest in the WHO European Region (19.8 per 1000) and lowest in the WHO Eastern Mediterranean Region (0.1 per 1000) [1]. Unfortunately, many women drink alcohol (EtOH) before they know they are pregnant, and this poses a potential diagnostic problem because FASDs cannot be diagnosed early in utero [1,2] when it might be easiest to intervene therapeutically. In addition, almost half of the estimated 80,000 children born with FASDs each year in the US go undiagnosed. Children with FASDs have facial abnormalities, small eyes and head size, and prominent cognitive and behavioral deficits [3]. The exact mechanisms for these abnormalities are not known, and the mechanisms by which EtOH disrupts fetal brain development are complex and incompletely understood, as are the genetic factors that modify individual vulnerability. The present study is part of a series aimed at identifying potential biomarkers for FASDs, with the goal of testing their predictive value in identifying which at-risk fetuses will present the clinical features of FASDs in post-natal life [4,5].

Since FASDs cannot be diagnosed early in utero by conventional imaging, there is a need to find early biomarkers that can predict which at-risk fetuses will go on to have FASDs postnatally and provide clues to early interventions that might prevent or ameliorate FASDs. Among the potential biomarkers, microRNAs (miRNAs) have attracted attention [6,7,8,9,10,11]. miRNAs act by binding to or destabilizing mRNAs and repressing protein translation and play important roles in brain development [12]. They also have been implicated in neurological diseases, including those caused by prenatal exposure to EtOH. Because individual miRNAs are involved in regulating expressions of hundreds of genes on average, they represent attractive subjects of investigation to elucidate the pathogenesis of complex disorders [13], such as FASDs. A summary of published data on some of the molecular targets of specific miRNAs and their associated neurodevelopmental functions or disorders are shown in Appendix A [14,15,16,17,18,19,20,21,22,23,24,25,26,27,28,29,30,31,32,33,34,35,36,37,38,39,40,41,42,43,44,45,46,47,48,49,50,51,52,53,54,55,56,57,58,59,60,61,62,63,64,65,66,67,68,69,70,71,72,73,74,75,76,77,78,79,80,81,82,83,84,85,86,87,88,89,90,91,92,93,94,95,96,97,98,99,100,101,102,103,104,105,106,107,108,109,110,111,112,113,114,115,116,117,118,119,120,121,122,123,124,125,126,127,128,129,130,131,132,133,134,135,136,137,138,139,140,141,142,143].

Changes in miRNA levels have been described in animal models of FASDs, as well as in human fetuses exposed to EtOH in vivo [144,145,146,147]. Thus, FASDs might be predicted by altered fetal brain levels of specific miRNAs, including microRNA-9 (miR-9) and its target molecules, e.g., brain-derived neurotrophic factor (BDNF), restrictive element-1 silencing transcription factor (REST), and Sonic hedgehog protein (Shh). Our laboratory has developed methods to measure the cargo of fetal brain-specific exosomes (FB-Es) that can be harvested non-invasively from the mother’s blood [148,149]. Correlation between FB-E miRNA changes and changes in an anatomical hallmark of FASDs, e.g., eye diameter [150,151], would suggest that specific miRNAs in FB-Es might be useful in the detection of FASDs early in utero and suggest possible therapeutic approaches aimed at preventing or ameliorating FASDs.

## 2. Results

### 2.1. Prenatal EtOH Exposure Was Associated with Reduced miRNA Levels in Fetal Brain but Increased miRNA Levels in Maternal Serum during Development

To understand the dynamics of miRNA changes, we quantified the expression levels of 84 miRNAs that are important for neurogenesis and neuronal development, including miR-9 and miR-132, in human fetal brain samples and matching maternal serum samples obtained from a previously organized human tissue biobank, created to study the effects of prenatal EtOH exposure on fetal brain development. miR-9 and miR-132 share molecular targets, and in rodent and ovine fetuses, had been shown to be downregulated on exposure to EtOH [147]. Each EtOH-exposed case was paired with a fetal sex-, GA-, maternal age-, and race-matched control from GA 9–23. Biometric and clinical characteristics of the subjects are shown in Table 1.

Table 1. Fetal brain and maternal blood used in miRNA studies. EtOH cases (n = 40) vs. Controls (n = 40). Ethanol-exposed fetal brain samples and blood samples from mothers who consumed EtOH during pregnancy were matched with non-EtOH-exposed controls by sex, GA, and race (white: Caucasian; black: African-American, some of them brown-skinned). In our population, other racial minorities were relatively few. Therefore, for this limited study, other non-Caucasian groups were not included but will be incorporated in future larger-scale studies, along with maternal age. PCR for the SRY gene was performed to determine fetal sex.

The 84 assayed miRNAs were represented on a Neurologic Disorders-Associated miRNA Array (Qiagen), which included markers previously reported to be associated with either normal neurodevelopment or neurological/neuropsychiatric disorders. Table 2 lists the miRNAs included in the array, together with the associated conditions that are pertinent to the neurodevelopment and neuropathology of FASDs. Major miRNAs implicated in CNS development and neurogenesis are presented in Table 3 [152,153,154,155,156,157,158,159,160,161,162,163,164,165,166,167,168,169,170,171,172,173,174,175,176,177,178,179,180,181,182,183].


**Codes**


NPCs: neural progenitor cellsELAVL3/4: embryonic lethal abnormal vision-like 3/4;EZH2: enhancer of zeste 2 polycomb repressive complex 2 subunit;MSI1: Musashi1PTBP1/2: polypyrimidine tract binding protein 1/2REST: RE1 Silencing Transcription FactorSCP1: CTD small phosphatase 1 (also known as CTDSP1)ZFP36: ZFP36 ring finger protein;ZFP36L1: ZFP36 ring finger protein like 1Sox9: SRY-Box Transcription Factor 9NOVA1: NOVA Alternative Splicing Regulator 1, Neuro-Oncological Ventral Antigen 1 N NOVA Alternative Splic1Rbfox1: RNA Binding Fox-1 Homolog 1Phf6: X-linked syndromic intellectual disability genePCM1: pericentriolar material 1AMPARs: AMPA-type glutamate receptorsCREB: cAMP response element binding proteinMeCP2: methyl CpG-binding protein 2Cdh5: vascular endothelial cadherin (VE-cadherin)FMRP: fragile X mental retardation protein

Figure 1A illustrates the developmental changes in overall miRNA expression in the maternal serum and fetal brain, presented as heat maps of the 84 miRNAs in the microarray, relative values at 11.3 weeks gestational age (GA), which is defined as baseline. Changes in the developmental pattern associated with exposure to EtOH are illustrated in Figure 1B. The position of each miRNA in the 96-well plate is specified in Figure 1C. Each sample was used for one 96-well pre-plated miRNA array. Thus, a total of 24 plates were used for 24 samples in the array: six control maternal sera, six EtOH-exposed maternal sera, six control fetal brains, and six EtOH-exposed fetal brains.

Compared to the 11.3 weeks baseline, miRNA expression in the unexposed control maternal serum was reduced (green) at 12.2 weeks GA and even lower at 18.3 weeks (Figure 1A, top). This pattern was less consistent in the fetal brain (mostly neocortex, Figure 1A, bottom). When pregnant women who consumed EtOH were compared to their individually matched, unexposed controls, the EtOH-exposed group showed greatly reduced (green) serum miRNA levels at 11.3 weeks GA, but then the levels increased (red) dramatically by 18.3 weeks (Figure 1B, top). The relative changes in miRNA levels in the fetal brains were less consistent, but generally, the levels were slightly lower in the EtOH-exposed fetal brains than in their individually matched controls.

### 2.2. EtOH Exposure Was Associated with Changes in Expression of Array miRNAs

EtOH exposure was associated with dysregulation of miRNA signaling in both the fetal brain and maternal serum. From a combined heatmap expression of several miRNAs appeared to be most dramatically changed. Nine of these miRNAs and their downstream targets are listed in Table 4 and were assayed by qRT-PCR in six EtOH-exposed and six unexposed matched control maternal sera and the corresponding fetal brain homogenates. All but one miRNA (mIR-509) showed significant changes in expression.

In the control cases, a general upregulation of miRNA levels was seen in the secondtrimester maternal serum, compared to the first trimester (Figure 2A). Prenatal exposure to EtOH was associated with a reduction of this upregulation from 3.8-fold to 2.0-fold (Figure 2A). By contrast, in EtOH-exposed fetal brains, the normal 1.9-fold downregulation of miRNA levels seen in controls was converted to a 2.0-fold increase late in the second trimester (Figure 2B). These responses differed by GA. In the first-trimester samples, screened miRNAs were upregulated 2.5–5.5-fold in the fetal brains and downregulated 3.5–6.5-fold in the maternal serum, compared to their matched unexposed controls. In the second-trimester samples, EtOH exposure was associated with smaller effects in the opposite direction–downregulation in the fetal brain specimens and slight upregulation in the maternal serum. Thus, there was a positive correlation between the paired maternal and fetal specimens for several miRNAs at early GAs but a smaller or even negative correlation at late GAs.

### 2.3. EtOH Exposure Was Associated with Reduced miR-9 and miR-132 in FB-Es

EtOH exposure was associated with reduced miR-9 and miR-132 in FB-Es. Fluctuations of miR-9 and miR-132 levels were analyzed in FB-Es, which carry fetal brain cell-specific nucleic acids, proteins, lipids, and metabolites. FB-Es are produced in the endosomal compartment of the fetal brain cells and released into the maternal blood. Their contents reflect the metabolic and functional state of their cells of origin. FB-Es isolated from the maternal serum were assayed by qRT-PCR for two EtOH-responsive miRNAs, miR-9 and miR-132, which are enriched in neural cells and have been implicated in the regulation of neurogenesis, axon extension, dendritic growth, synaptic structure and function, vascular integrity, and microglial homeostasis [164,172,173,174,175,176,177,178,179,180,181,182,183]. A larger number of cases was analyzed by limiting the control-matching to GA and fetal sex only. Twenty EtOH-exposed cases were compared with their individually matched unexposed controls, as well as with non-pregnant controls. In these same cases, both miR-9 (Figure 3A) and miR-132 (Figure 3B) were downregulated in FB-Es from pregnant mothers who consumed EtOH.

### 2.4. miR-9 Expression in FB-Es

Copy numbers of miR-9 were measured in FB-Es from the first 10 available pairs of EtOH-exposed mothers and their GA- and fetal sex-matched controls using digital droplet PCR (ddPCR; Figure 4). In each case, the EtOH-exposed sample had fewer copies than its control. The miR-9 levels in the control FB-Es ranged from 200 to 1200 copies per 1 μL, while 180 to 820 copies were found in the EtOH-exposed FB-Es. With the accumulation of additional cases, it was possible to add subjects for other assays reported below.

### 2.5. miRNA-9 Targets in a Double Negative Feedback Loop Pathway

In both the developing and adult vertebrate brains, miR-9 is expressed at high levels and is involved in regulating the proliferation of neural progenitors [137]. It also is important in the regulation of axon extension and local branching by targeting BDNF, which affects critical components of the cytoskeleton [184]. To understand whether EtOH exposure affects not only miR-9 but also its targets, we performed ELISA and quantitative Western-blot assays (qWestern) on the FB-E cargo from 20 pregnant women and their individually matched controls. There was a 50% reduction in levels of Synapsin (↓2.1-fold; *p* < 0.01; Figure 5A). Similarly, downregulation was observed in the miR-9 downstream double-negative feedback targets REST (↓1.8-fold; *p* < 0.05; Figure 5B), Shh (↓2.2-fold; *p* < 0.05; Figure 5C), and BDNF (↓1.4-fold; *p* < 0.05; Figure 5D), suggesting that EtOH exposure might be associated with synaptic injury and that this could be detected non-invasively using FB-Es. The postulated double-negative feedback loop between miR-9 and its molecular target REST and a triple-negative net positive feedback loop through BDNF were previously shown by others [184,185,186,187,188,189] and are diagrammed in Figure 5E. The net effect of these feedback loops is that miR-9 controls its own expression via the regulation of its targets.

### 2.6. Reductions in FB-E miR-9 Levels in Fetuses Exposed to EtOH Correlate with Reductions in Eye Diameter in Fetuses Exposed to EtOH in a Larger Population

Previously, we measured eye diameters in 10 histological sections of human fetuses that had or had not been exposed to EtOH. Levels of miR-9 were measured by ddPCR in FB-Es isolated from the maternal blood drawn at the time of voluntary pregnancy termination. A linear correlation was observed between the reduction in eye size (arithmetic difference between the EtOH-exposed fetus and its paired control) and the reduction in exosomal miR-9 levels. To determine whether FB-E miR-9 levels might be generally useful as a molecular marker to identify those at-risk fetuses that are destined to be born with FASDs, we performed ddPCR assays for FB-E miR-9 in a larger number of cases (N = 40 EtOH-exposed cases and their individually GA-matched unexposed controls), achieved by eliminating maternal age, maternal race, and fetal sex as control variables. The correlations persisted in this larger population (Figure 6). The first-trimester pregnancies ranged from 9 to 14 weeks GA, and the second-trimester pregnancies ranged from 14.1 to 23 weeks GA. The differences in FB-E miR-9 levels between the EtOH and control groups were highly significant (*p* = 0.000000464825; Figure 6A). *** *p* < 0.001. Each assay was performed in triplicate, and the values indicated by the dots are the averages of the three determinations. The levels of miR-9 in the control FB-Es rose dramatically early in the second trimester, whereas the miR-9 levels of the EtOH-exposed group did not. Each EtOH-exposed case is compared with its individual control in Figure 6B. Except for the two youngest fetuses, miR-9 levels were lower in each EtOH-exposed case than in its control. Correlations between the reduction in eye size and the reduction in exosomal miR-9 levels are presented as a scatter plot in (Figure 6C). As in Figure 6, the correlation appeared to be almost constant throughout the observed gestational periods when the data were presented as % change (Figure 6D).

## 3. Discussion

This preliminary study suggests that miRNAs are important cellular targets of fetal alcohol exposure that can be detected early in gestation, even in the first trimester. In particular, miR-9 is identified as a possible molecular biomarker for FASDs that can be quantified in FB-Es isolated non-invasively from maternal blood. These two findings may allow us to conduct large-scale population-based studies to determine whether FB-E miR-9 levels can be used to predict the emergence of FASDs postnatally. Compared with the controls, subjects with EtOH exposure in the first and/or second trimester showed changes in miRNA levels in both the fetal brain and maternal serum. In the first trimester samples, screened miRNAs were downregulated in the maternal serum but upregulated in the fetal brain. In the early second trimester samples, there was a positive correlation between the paired maternal and fetal specimens for several miRNAs, but in the late second trimester, the correlation was much less, or even negative. Because the changes in the fetal brain did not mirror the changes in maternal blood, the effect of EtOH exposure on the fetal brain cannot be attributed to the passive transmission of maternal blood miRNAs to the fetus. The early emergence of changes in molecular markers is important because they might suggest potential therapeutic approaches to limiting the negative effects of exposure to EtOH soon after a woman learns she is pregnant and realizes that her fetus is at risk.

### 3.1. MicroRNAs as Indicators of Alcohol-Associated Fetal Pathology

The effects of prenatal EtOH exposure on fetal development are complex and often severe. Abnormalities include prenatal and postnatal growth retardation, CNS injury, and facial abnormalities [3,190,191,192]. These effects are difficult to detect early in pregnancy with the current imaging technology. Thus, it is important to develop novel non-invasive tools to predict FASDs in utero. The present study focused on a molecular biomarker, miR-9, which is highly expressed in the fetal brain, can be detected early in fetal development, and is important in regulating axonal elongation and branching via its reciprocal interactions with BDNF and consequent effects on the cytoskeleton. miR-9 also is involved in regulating the proliferation of neural progenitors.

By use of qRT-PCR arrays and even more sensitive ddPCR, we have demonstrated the following: (i) the developmental regulation of miRNAs in the human fetal brain; (ii) the effects of in utero EtOH exposure on miRNA levels, both in the developing brain and in FB-Es; (iii) exposure to EtOH reduces miR-9 levels in FB-Es; and (iv) exposure to EtOH is associated with reduced levels of miR-9 target proteins. It has been suggested that the toxic effects of EtOH on miRNA expression occur primarily during the second trimester, when neurogenesis is active [191]. Interestingly, the numbers of EtOH-sensitive miRNAs increased during the neural differentiation stage, from 4% in fetal neural stem cells (NSCs) to 11% in differentiated neurons, due to the increasing complexity of miRNA contents during neuronal maturation [193,194]. Also, the EtOH-sensitive miRNAs miR-335 and miR-21, which play important roles in regulating NSC fate [194], are no longer EtOH-sensitive by the end of the second trimester. Other EtOH-sensitive miRNAs, such as miR-10a/10b, negative regulators of the Hox gene family, can dysregulate neuronal migration when upregulated during a critical window for neuronal migration, while miR-21 and miR-335 regulate NSC behavior at earlier times [193,195]. Several miRNAs, including miR-9, are EtOH-sensitive during multiple developmental periods (from the embryonic and fetal stages to adulthood [195,196,197,198]). EtOH reduced miR-9 expression early in mouse and fish development [196,198,199] and increased miR-9 expression at later stages of fetal development and in adult animals [195,197]. This developmental switch is probably due to the effects on miR-9 targets, but the exact mechanisms are not completely understood. In the present study, we looked not only at the expression of miR-9 but also its targets (BDNF, Shh, Synapsin, and REST), which are affected by EtOH via its action on miR-9 throughout development. It is likely that downstream effects on the translation of those proteins would change during neuronal maturation and into adulthood. By assessing miR-9 in FB-Es, we avoided confusion caused by the mixing of fetal and maternal miR-9 and its targets.

### 3.2. FB-Es as Non-Invasive Tools to Assess Potential Molecular Biomarkers

In previous studies, we showed that fetal brain-derived exosomes (FB-Es) can be isolated non-invasively from the maternal blood and contain neuron- [148] and oligodendrocyte-specific [149,150,151] molecules that might be useful as biomarkers for the diagnosis of pathological conditions. Some of these markers were associated with anatomical abnormalities characteristic of FASDs, including reduced eye size. In the present study, we determined how exposure to EtOH during pregnancy affects miRNA levels in both the fetal brain and maternal serum. Based on the results of previous microarray analysis (not published), we focused attention on whether FASDs can be predicted by altered levels of miR-9 in fetal brain-derived exosomes (FB-Es) isolated non-invasively from maternal serum. We tested whether the results from fetal brains and FB-Es were congruent and whether any observed abnormalities are associated with the effects of EtOH on eye diameter. This anatomical hallmark was used because eyes appear early in embryogenesis and because EtOH-associated reductions of eye diameter have been detected even in the first trimester.

The association between early exposure to EtOH and changes in FB-E miRNA levels was detected long before facial features of FASDs could be seen by conventional ultrasound imaging or even MRI. Of particular importance, the EtOH-associated reduction in miR-9 levels, not only in the fetal brains but also in FB-Es, correlated with the reduction in fetal eye diameters, a common morphological feature of FASDs [3]. A similar correlation had been found previously between FB-E myelin basic protein (MBP) levels and reduced eye diameter [150].

### 3.3. The Association between Exposure to EtOH and Reduced Fetal miRNA Expression Probably Is Due to a Direct Effect of EtOH on Fetal Brain

EtOH exposure was associated with the upregulation of target miRNAs in maternal serum during the second trimester but with downregulation during this time in the fetal brain. Therefore, the effects of EtOH on fetal brain miRNA expression are unlikely to result passively from changes in maternal blood concentrations. Exposure to EtOH was associated with changes in the expression of eight neurodevelopment- and neurological disorder-related miRNAs in maternal serum and fetal brain homogenates. A general downregulation seen in the first trimester disappeared later in development, suggesting that if miRNAs were to be used as biomarkers for FASDs, they would have to be assayed during the first trimester of pregnancy, when imaging methods are least able to resolve the anatomical abnormalities associated with FASDs.

### 3.4. Limitations

The present study should be viewed as preliminary, in part because the number of EtOH-exposed cases and controls was too small to permit control for all the potential variables that might affect fetal sensitivity to EtOH, such as EtOH dose, maternal obesity, use of tobacco and other drugs of abuse, medications, and socioeconomic status. An ethically unavoidable limitation is that the study was not prospectively randomized, i.e., although we tried to pair each EtOH-exposed fetus with an unexposed control, women were not randomly assigned prospectively to either use EtOH or not use EtOH. Because the biobank from which the samples were obtained was not collected solely to study FASDs, it did not include tissue specimens for biochemical testing to verify alcohol use. However, any under-reporting of EtOH use would have the effect of reducing rather than exaggerating the reported effects of EtOH exposure. Moreover, in studying some other drugs of abuse in our patient population, there was a very high correlation between self-report and drug test results. Finally, since the pregnancies all were terminated, we cannot say for certain which fetuses would have gone on to develop FASDs postnatally. We now are conducting a larger prospective study of pregnancies brought to term (including tissue ethylene glycol testing to verify EtOH use), so that we can determine which at-risk children develop FASDs, and possibly identify molecular determinants of the subtypes and severities of the FASD. Detailed studies on other miRNAs, e.g., miR-124, will be included in future publications.

## 4. Materials and Methods

### 4.1. Clinical Samples

Pregnant women who used EtOH were compared with those who did not use alcohol during pregnancy and did not use any drugs or medications (Table 1). Cases were selected based on the availability of fetal brain and eye tissues, matching maternal blood samples, and, at a minimum, on data for fetal sex and GA. In some cases, the controls also were matched with regard to maternal race. Each EtOH-exposed fetus was paired with a sex- and GA-matched control. Consenting women were enrolled between 9- and 23-week GA under Temple University Institutional Review Board (IRB)-approved protocol. Data for both sexes were combined in all assays.

Subject Recruitment. Pregnant women with or without EtOH use during pregnancy were grouped in two GA windows within the first or second trimester (9–23 weeks). GA was confirmed by an ultrasound performed prior to recruitment. Samples from 40 women with or without EtOH use were collected.

Assessment of EtOH Exposure in Pregnancy. Maternal EtOH exposure was determined using a face-to-face questionnaire. Exposure status was based on self-reported EtOH use (modified timeline follow-back). Pregnant women were screened for EtOH use, and then maternal blood, fetal brain (mostly cortex), and eye tissue were collected.

The amount of EtOH was calculated as the total number of drinks consumed in a week multiplied by the number of weeks of exposure. A detailed questionnaire was used based on the NICHD PASS study [200]. Each drink was estimated as the equivalent of one shot (1.5 oz of brandy or 5 oz of wine [148].

Tissue collection. Fetal brain and eye tissues and maternal blood from subjects undergoing elective termination of pregnancy were collected according to an IRB-approved protocol by a trained study coordinator and were transferred to the laboratory within 60 min.

### 4.2. RNA Preparation

Total RNA was isolated from the fetal brain and FB-Es using the RNeasy Kit (Qiagen, Valencia, CA, USA) with on-column DNA digestion. Fetal sex determination was performed using SuperScript One-Step RT-PCR with Platinum Taq (Life Technologies, Carlsbad, CA, USA), BioRad C1000 Touch Thermal Cycler, and sex-determining SRY primers. The qRT-PCR reaction was performed using the One-Step FAST SYBR Green PCR Master Mix (Qiagen, Valencia, CA, USA). For relative quantification, the expression level of genes was normalized to the housekeeping gene β-actin.

### 4.3. miRNA Preparation and Real-Time qRT-PCR

Human fetal brain and maternal plasma total miRNA/RNA was isolated with the miRNeasy kit (Qiagen, Valencia, CA, USA), using QIAzol Lysis Reagent with on-column DNA digestion. The qRT-PCR reaction was performed with 1 μg total RNA/miRNA using the One-Step FAST RT-PCR SYBR Green PCR Master Mix (Qiagen). The StepOnePlus Real-Time PCR system thermocycler was used (Applied Biosystems, Waltham, MA, USA). PCR conditions were as follows: activation 95 °C 5 min, PCR 45 cycles: 95 °C 10 s, 60 °C 20 s, 72 °C 30 s, melting curve (95–65 °C), cool to 40 °C 30 s. For the relative quantification, the expression levels of genes were normalized to the housekeeping gene β-actin.

cDNA was obtained using the miRNAs and miScript PCR Systems, containing 5× miScript HighSpec Buffer, 10× miScript Nucleics Mix, and miRNA Reverse Transcriptase mix (Qiagen, Valencia, CA, USA). miRNA Neurological Development and Disease Pathway array (Qiagen) for 90 miRNAs (Table 2, Table 3 and Appendix A) was assayed by real-time PCR with the Applied Biosciences Cycler using the fetal brain RNAs, primers, and Cyber Green mix (Qiagen Universal PCR Master Mix). Of the 90 miRNAs on the array, 6 were controls that enabled data analysis using the ΔΔCT method of relative quantification, assessment of reverse transcription performance, and assessment of PCR performance. Real-time PCR experiments were performed on an Applied Biosystems instrument with the following thermal-cycling procedure: 95 °C for 10 min, followed by 40 cycles of 95 °C for 15 s, and 56 °C for 1 min, as specified in the RNA assay protocol provided by Applied Biosciences.

### 4.4. Droplet Digital PCR (ddPCR)

For the absolute quantitation of mRNA copies, ddPCR was performed using the QX200 Droplet Digital PCR (ddPCR) System (Bio-Rad Laboratories, Inc., Hercules, CA, USA) with QuantaSoft Analysis Pro Software v 1.4 (AP) (Bio-Rad, Hercules, CA, USA). Fifty nanograms of human fetal total RNA was used with the 1st Strand cDNA Synthesis Kit (Qiagen, Valencia, CA, USA). The cDNA (300 dilution) aliquots were added to the BioRad master mix to conduct ddPCR (EvaGreen ddPCR Supermix, BioRad, Hercules, CA, USA). The prepared ddPCR master mix for each sample (20-μL aliquots) was used for droplet formation. PCR conditions were as follows: activation at 95 °C 5 min, PCR 45 cycles at 95 °C 10 s, 60 °C 20 s, 72 °C 30 s, melting curve (95–65 °C), cool to 40 °C 30 s. The absolute quantity of DNA per sample (copies/µL) was calculated using QuantaSoft Analysis Pro Software v 1.4 (AP) (Bio-Rad, Hercules, CA, USA) to analyze ddPCR data for technical errors (Poisson errors). With 20,000 droplets, the above ddPCR protocol yields a linear dynamic range of detection between 1 and 100,000 target mRNA copies/µL. The ddPCR data were exported to Microsoft Excel (Microsoft 365) for further statistical analysis.

### 4.5. Primers (IDT Inc., Coralville, IA, USA)

β-actin: S: 5′-CTACAATGAGCTGCG TGTGGC-3′,

AS: 5′-CAGGTCCAGACGCAGGATGGC-3′,

BDNF: S: 5′-CAGGGGCATAGACAAAAG-3′, AS: 5′-CTTCCCCTTTTAATGGTC-3′,

AS: 5′-GAGGGACTGAGCTGGACAACCCAC-3′

SRY: Forward 5′-CAT GAA CGC ATT CAT CGT GTG GTC-3′; reverse 5′-CTG CGG GAA GCA AAC TGC AAT TCT T-3′.

Hsa-mir-26b MI0000084

5′CCGGGACCCAGUUCAAGUAAUUCAGGAUAGGUUGUGUGCUGUCCAGCCUGUUCUCCAUUACUUGGCUCGGGGACCGG-3′

Hsa-mir-132 MI0000449

5′CCGCCCCCGCGUCUCCAGGGCAACCGUGGCUUUCGAUUGUUACUGUGGGAACUGGAGGUAACAGUCUACAGCCAUGGUCGCCCCGCAGCACGCCCACGCGC-3′

Hsa-mir-124-1 MI0000443

5′AGGCCUCUCUCUCCGUGUUCACAGCGGACCUUGAUUUAAAUGUCCAUACAAUUAAGGCACGCGGUGAAUGCCAAGAAUGGGGCUG-3′

Hsa-mir-125b-1 MI0000446

5′UGCGCUCCUCUCAGUCCCUGAGACCCUAACUUGUGAUGUUUACCGUUUAAAUCCACGGGUUAGGCUCUUGGGAGCUGCGAGUCGUGCU-3′

Hsa-mir-138-1 MI0000476

5′CCCUGGCAUGGUGUGGUGGGGCAGCUGGUGUUGUGAAUCAGGCCGUUGCCAAUCAGAGAACGGCUACUUCACAACACCAGGGCCACACCACACUACAGG-3′

Hsa-mir-128-1 MI0000447

5′UGAGCUGUUGGAUUCGGGGCCGUAGCACUGUCUGAGAGGUUUACAUUUCUCACAGUGAACCGGUCUCUUUUUCAGCUGCUUC-3′

Hsa-mir-509-1 MI0003196

5′CAUGCUGUGUGUGGUACCCUACUGCAGACAGUGGCAAUCAUGUAUAAUUAAAAAUGAUUGGUACGUCUGUGGGUAGAGUACUGCAUGACACAUG-3′

Hsa-mir-9-1 MI0000466

5′CGGGGUUGGUUGUUAUCUUUGGUUAUCUAGCUGUAUGAGUGGUGUGGAGUCUUCAUAAAGCUAGAUAACCGAAAGUAAAAAUAACCCCA-3′

Hsa-mir-134 MI0000474

5′CAGGGUGUGUGACUGGUUGACCAGAGGGGCAUGCACUGUGUUCACCCUGUGGGCCACCUAGUCACCAACCCUC-3′

Hsa-mir-485 MI0002469

5′ACUUGGAGAGAGGCUGGCCGUGAUGAAUUCGsAUUCAUCAAAGCGAGUCAUACACGGCUCUCCUCUCUUUUAGU-3′

SNORD61-11

5′GCTATGATGAATTTGATTGCATTGATCGTCTGACATGATAATGTATTTTTGTCCTCTAAGAAGTTCTGAGCTT-3′

### 4.6. ELISA Quantification of Exosomal Proteins

Proteins and CD81 (American Research Products-Cusabio) were quantified by human-specific ELISAs according to the suppliers’ instructions. ELISA data were statistically evaluated using Excel (Microsoft 365, software version 2404) and statistical analysis tools: CurveExpert for ELISA statistics (CUSABIO) or APP 96-well Plate Assay Data Analysis Software 5.0.apk (Cloud-Clone, Katy, TX, USA), available online.

### 4.7. Quantitative Western Blots

Thirty micrograms of proteins in Laemmli sample buffer were heated at 95 °C for 10 min and separated by 10% sodium dodecyl sulfate-polyacrylamide gel electrophoresis (SDS-PAGE), then transferred to supported nitrocellulose membranes. Primary antibodies: anti-REST rabbit polyclonal (AB10514P, Millipore, Burlington, MA, USA), anti-Synapsin, and anti-α-tubulin clone B512 (Sigma-Aldrich, St. Louis, MO, USA). Secondary IRDye^®^ antibodies were used to detect band intensity (normalized to tubulin) using an Odyssey^®^ CLx Imaging System (iS Image Studio™ Software version 3.1) and LiCor dyes.

### 4.8. Isolation of Fetal Brain-Derived Exosomes (FB-Es) from Maternal Serum and ELISA Quantification of Exosomal Proteins

Human FB-Es were isolated as described previously [148]. In brief, two hundred and fifty microliters of serum was incubated with exosome precipitation solution (EXOQ; System Biosciences, Inc., Mountainview, CA, USA). The resultant suspensions were centrifuged at 1500× *g* for 30 min at 4 °C, and pellets were resuspended in 400 mL of distilled water with protease and phosphatase inhibitor cocktail for immunochemical enrichment of exosomes. To isolate exosomes from fetal neural sources, total exosome suspensions were incubated for 90 min at 20 °C with 50 μL of 3% bovine serum albumin (BSA; Thermo Scientific, Inc., Waltham, MA, USA) containing 2 μg of mouse monoclonal IgG1 antihuman contactin-2/TAG1 antibody (clone 372913, R&D Systems, Inc., Minneapolis, MN, USA) that had been biotinylated (EZLink sulfo-NHS-biotin System, Thermo Scientific, Inc., USA). Then, 10 μL of Streptavidin-Plus UltraLink resin (PierceThermo Scientific Inc., Waltham, MA, USA) in 40 μL of 3% BSA was added, and the incubation continued for 60 min at 20 °C. After centrifugation at 300× *g* for 10 min at 4 °C and removal of supernatants, pellets were resuspended in 75 μL of 0.05 mol/L glycine-HCl (pH 3.0), incubated at 4 °C for 10 min, and recentrifuged at 4000× *g* for 10 min at 4 °C. Each supernatant was mixed in a new 1.5 mL Eppendorf tube with 5 mL of 1 mol/L Tris-HCl (pH 8.0) and 20 μL of 3% BSA, followed by the addition of 0.4 mL of mammalian protein extraction reagent (M-PER; Thermo Scientific Inc., Waltham, MA, USA) containing protease and phosphatase inhibitors prior to storage at −80 °C. For the exosome counts, immunoprecipitated pellets were resuspended in 0.25 mL of 0.05 mol/L glycine-HCl (pH 3.0) at 4 °C with a pH of 7.0 and 1 mol/L Tris-HCl (pH 8.6). The exosome suspensions were diluted 1:200 to permit counting in the range of 1–5 × 10^8^/mL with an NS500 nanoparticle tracking system (NanoSight, Amesbury, UK).

### 4.9. Statistical Analysis

For the statistical analysis, we used SPSS Statistics from IBM Corp., released in 2017 for Windows, Version 25.0 (Armonk, NY, USA). All data are represented as the mean ± SD for all performed repetitions. Means were analyzed by a one-way ANOVA, with Bonferroni correction. Statistical significance was defined as *p* < 0.05. Sample numbers are indicated in the figure legends. Data from ddPCR, which measures absolute quantities of DNA per sample (copies/µL), were processed using QuantaSoft Analysis Pro Software (Bio-Rad) to analyze for technical errors (Poisson errors) and then exported to EXCEL for further statistical analysis.

### 4.10. Ethics: Human Subjects

Consenting mothers were enrolled between 9–23 weeks gestation under a protocol approved by Temple University Institutional Review Board (IRB). This protocol involved no invasive procedures other than routine care. Maternal EtOH exposure was determined with a face-to-face questionnaire that also included questions regarding many types of drugs/medications used. The questionnaire was adapted from that designed to identify and quantify maternal EtOH exposure in the NIH/NIAAA Prenatal Alcohol and SIDS and Stillbirth (PASS) study [199].

All procedures for collection and processing of human brain tissues and blood were performed according to NIH Guidelines by a trained Study Coordinator. All investigators completed Citi Program-Human Subject training, Blood-Borne Pathogens Training, and Biohazard Waste Safety Training annually.

Written informed consent has been obtained from the parents of the patient(s) for studies, and deidentified samples were used for this publication. Informed Consent forms were maintained by the Study Coordinator. The de-identified log sheets contain an assigned accession number, and the age, sex, ethnicity, and race of the patient. Except for an assigned accession number, no identification was kept on the blood samples.

a. Eligibility Criteria: The blood and tissue samples were obtained according to NIH Guidelines through a trained Study Coordinator. Samples were collected regardless of sex, ethnic background, and race.

b. Treatment Plan: Each patient was asked to sign a separate consent form for research on blood and tissue samples. Blood obtained was processed for collection of serum and plasma. No invasive procedures were performed on the mother, other than those used in her routine medical care. Fetal tissues were processed for RNA or protein isolation.

c. Risk and Benefits: There are very small risks of loss of privacy as with any research study where protected health information is viewed. The samples were depersonalized before they were sent to the lab for analysis. There were no additional risks of blood sampling as this was only performed in patients with clinically indicated venous access. There was little anticipated risk from obtaining approximately 2–3 cc of blood, but a well-trained Study Coordinator collected all samples.

There was no direct benefit to the research subjects from participation, but there is significant potential benefit for the future FASDs subjects and the general population. This research represents a reasonable opportunity to further the understanding, prevention, or alleviation of a serious problem affecting the health or welfare of FASDs patients.

d. Informed Consent: Consent forms were maintained by the Study Coordinator and were not sent to the investigator with the samples. The de-identified log sheets and IRB protocol were sent by the Study Coordinator to Principal Investigator with each blood and tissue sample. This sheet contains an assigned accession number, the age, sex, ethnicity, and race of the patient. Except for an assigned accession number, no identification was kept on the blood and tissue samples.

## 5. Conclusions

Many women drink EtOH before they know they are pregnant and discontinue drinking once their pregnancy is diagnosed. Not all fetuses that are exposed to EtOH develop FASDs, but EtOH-associated reductions in fetal eye diameter occur even in the first trimester. Unfortunately, this and other somatic features of FASDs cannot be detected early in gestation noninvasively by conventional imaging. Thus, it is important to find early molecular biomarkers to predict which at-risk fetuses will go on to develop FASDs. The present study proposes a novel class of biomarkers based on miRNAs that are developmentally regulated and can be detected as early as 9 weeks of gestation. In the present study, EtOH exposure was associated with reduced expression of miR-9 and of its target proteins, BDNF, REST, Shh, and Synapsin, in FB-Es. This may allow biomarkers for FASDs to be assayed non-invasively from a small sample of maternal blood. The biomarkers so identified also may provide hints to therapeutic approaches that could prevent or ameliorate FASDs.

## Figures and Tables

**Figure 1 ijms-25-05826-f001:**
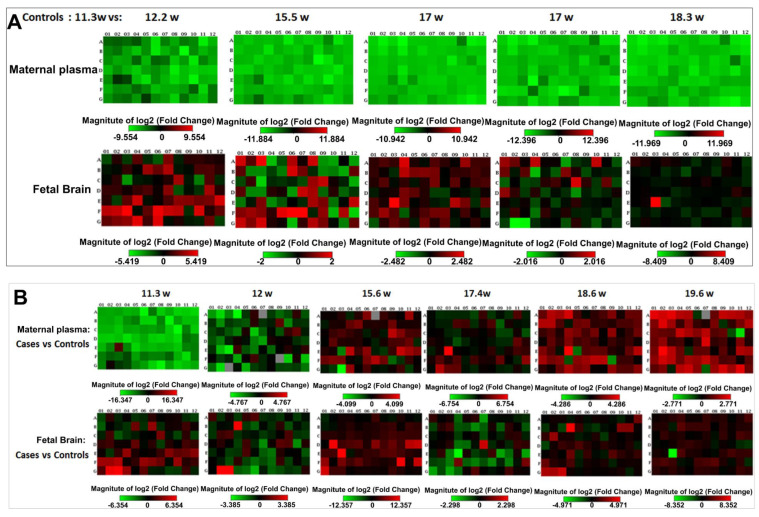
Effects of EtOH exposure on the normal reductions in miRNA expressions in fetal brain and maternal plasma during the course of gestation. Non-EtOH-exposed human fetal brain tissues from the first and second trimesters were compared with unexposed controls with respect to the expression of 84 miRNAs, using Real-Time qRT-PCR. The results in fold-changes are displayed as heat maps. To assess the normal developmental changes, in (**A**), only unexposed controls were measured and expressed as fold-changes relative to their expression levels at the earliest-studied time point, 11.3 weeks GA, which is not shown because, by definition, the values equal 1. Expression of the selected miRNAs generally decreased (green) at later GAs in maternal blood (top). The pattern was less clear in the fetal brain (bottom), with expression levels at first increasing (red) but then decreasing toward the 11.3 values (black). N = 6 for each group. In (**B**), each box represents the average change in expression for a specific miRNA in EtOH-exposed cases relative to their unexposed individually matched controls. N = 6 paired cases for each group, including the 11.3-week time point in (**A**). In the second trimester, EtOH exposure was associated with a significant upregulation (red) of target miRNAs in the maternal blood (top) but a downregulation (green) in the fetal brain (bottom). (**C**) The position of each studied miRNA and housekeeping control gene in the 96-well array is shown (all controls and SNORDs for normalization were positioned in row H from H1 to H12). hsa: human, *Homo sapiens*; miR-9-3p vs. miR-9-5p: The 5p strand is present in the forward (5′-3′) sense, while the 3p strand is the reverse sense complementary strand.

**Figure 2 ijms-25-05826-f002:**
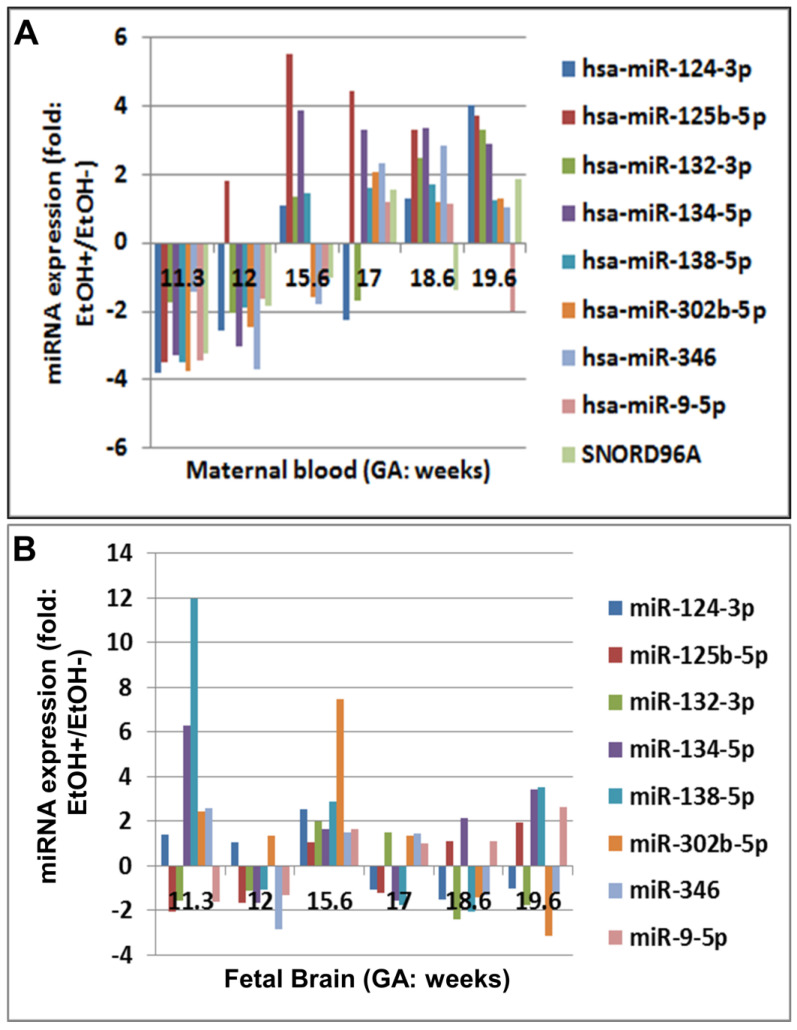
Exposure to EtOH is associated with changes in the expression of eight neurological disorder-related miRNAs. Expressions of candidate miRNAs from Table 4 were assayed by qRT-PCR on samples of maternal serum (**A**) and fetal brain homogenates (**B**) from 12 pregnancies and their fetal GA- and sex-matched controls. Results were expressed as fold change between each EtOH-exposed case and its control. An overall upregulation in miRNA expression was seen in the second trimester compared to the first in maternal serum but not in the fetal brain. The expression of miR-509 was almost unaltered, so it is not included in this graph.

**Figure 3 ijms-25-05826-f003:**
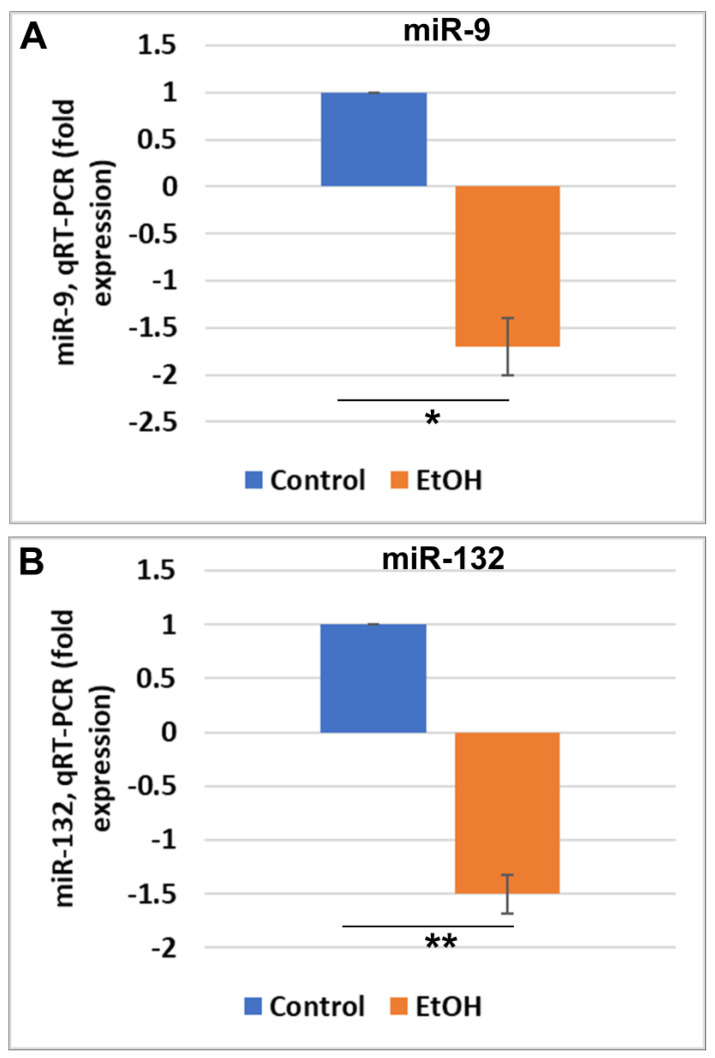
Exposure to EtOH is associated with reduced expression of miR-9 and miR-132 in FB-Es. (**A**) Levels of miRNA were measured by qRT-PCR in FB-Es of 20 EtOH-exposed maternal plasmas and compared with 20 individually matched unexposed controls, 10 matched pairs from the first trimester and 10 from the second. Expressions of miR-9 (**A**) and miR-132 (**B**) were normalized to SNORD, according to miScript Qiagen recommendations. Each assay was performed in triplicate and averaged. Changes in EtOH-exposed cases were expressed in comparison with their fetal sex- and GA-matched controls, measured relative to SNORD, and then their averages were assigned a value of 1. Error bars indicate the standard deviations for the 20 averages per group. Significance levels are for the comparison between all EtOH-exposed and all unexposed controls, based on ANOVA. * *p* < 0.05, ** *p* < 0.01.

**Figure 4 ijms-25-05826-f004:**
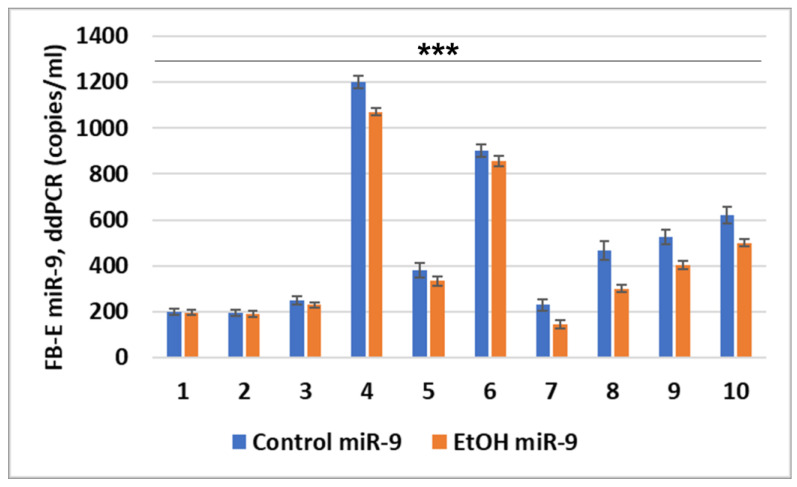
Prenatal exposure to EtOH is associated with inhibition of miR-9 expression in FB-Es. Absolute quantification of miR-9 in FB-Es isolated from the blood of 10 EtOH-exposed pregnant women and their individually matched controls was determined by measuring RNA copy numbers using ddPCR. Five ng of exosomal RNA (including miRNAs) was used in the reaction. Each assay was performed in triplicate. Control FB-Es contained 200–1200 copies of miR-9 per μL, while in EtOH-exposed FB-Es, 180–820 copies were found. In each case, the EtOH-exposed sample contained fewer copies of miR-9 than its GA-and sex-matched control (*** *p* = 0.006).

**Figure 5 ijms-25-05826-f005:**
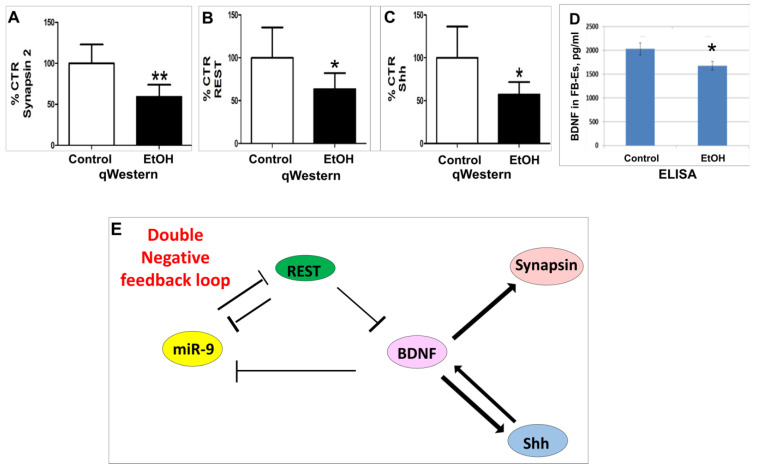
Potential targets of miRNA-9 in double-negative feedback loop pathways observed in FB-Es. The cargo of FB-Es from 20 pregnant women who had consumed EtOH and their individually matched controls were analyzed by ELISA and quantitative Western blot analysis (qWestern; (**A**–**C**)). Relative downregulation was observed quantitatively by qWestern for Synapsin (**A**), REST (**B**), and Shh (**C**) and by ELISA for BDNF (**D**). * *p* < 0.05, ** *p* < 0.01. (**E**). Proposed double-negative feedback loop pathways for miR-9 expression. In the brain, miR-9 inhibits transcription of the transcription factor REST, which in turn inhibits the expression of miR-9. REST also inhibits the translation of BDNF, which in turn inhibits the expression of miR-9, as well as upregulating Synapsin and reciprocally upregulating Shh. Activation of the Shh pathway induces an increase in BDNF expression and results in neuroprotection to oxidative stress.

**Figure 6 ijms-25-05826-f006:**
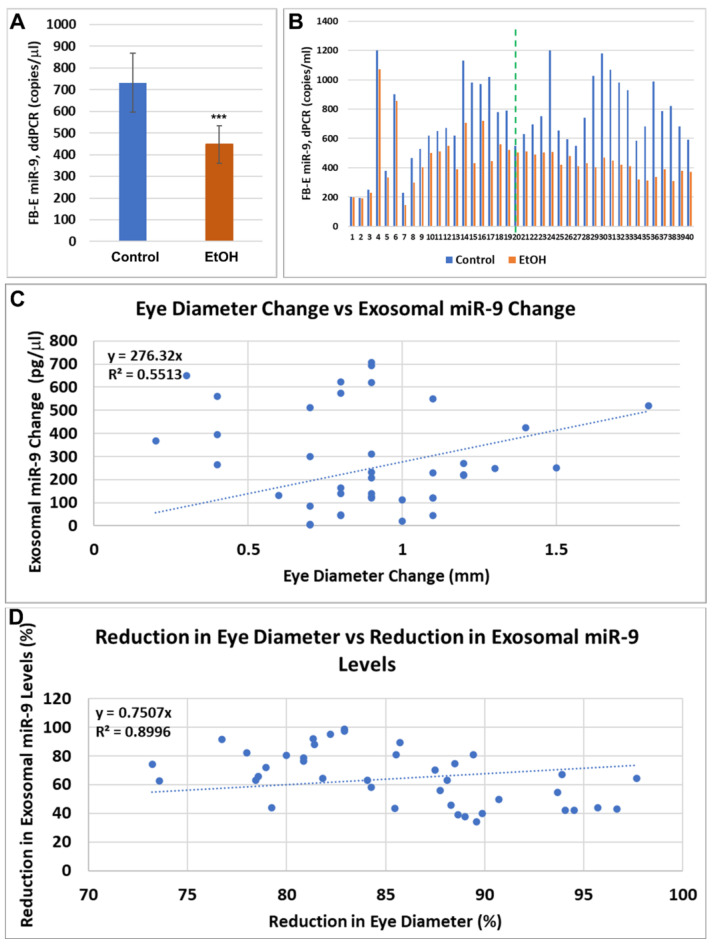
Reductions in FB-E miR-9 levels correlate with the reductions in eye diameter in a larger sample of fetuses exposed to EtOH. To enhance the potential practicality of FB-E miR-9 levels as a biomarker to predict FASDs, FB-Es were isolated from the blood of a larger group of mothers who consumed alcohol during pregnancy and their unexposed controls, matched only for GA from 9–23 weeks, disregarding other control factors, such as fetal sex. FB-E miR-9 levels were measured by ddPCR. Eye diameters were measured in histological sections of the fetuses. (**A**). Copy numbers of miR-9 were reduced by almost 37.5% in EtOH-exposed cases. (**B**). Each individual FB-E miR-9 level from the 40 EtOH-exposed fetuses is graphed next to its GA-matched control and arrayed in order of GA. In all but the two youngest cases, the EtOH-exposed levels were higher than those of their GA-matched controls. Note also that the control levels of miR-9 rose dramatically early in the second trimester, whereas the miR-9 levels of the EtOH-exposed group did not show this increase. The assay was performed in triplicate. The dashed green line separates the first-trimester cases (on the left) from the second-trimester cases (on the right). (**C**). The correlation between the reduction in eye size (difference between EtOH-exposed fetus and its paired control) and the reduction in exosomal miR-9 levels is presented as a scatter plot. First-trimester (9 to 14 weeks GA) and second-trimester pregnancies (14.1 to 23 weeks GA) are graphed together. *** *p* << 0.001 (*p* = 0.000000464825), based on Spearman’s correlation with exact two-tailed critical *p* values. Data in (**D**) are presented in %.

**Table 1 ijms-25-05826-t001:** Clinical characteristics of subjects used in the miRNA experiments.

	EtOH-Consuming Subjects (n = 40)	Control Subjects (No EtOH, n = 40)
Maternal Age (years ± SD)	26.17 ± 2.15	22.34 ± 1.70
Gestational Age (weeks ± SD)	15.47 ± 1.33	15.16 ± 1.42
Race: White vs. Black (%)	50 vs. 50	50 vs. 50
Fetal Sex (male vs. female, %)	50 vs. 50	50 vs. 50

**Table 2 ijms-25-05826-t002:** miRNAs in the Neurological Development and Disease Array.

miRNAs in Neurological Development and Disease

***Development***: miR-124-3p, miR-125b-5p, miR-132-3p, miR-134, miR-138-5p, miR-9-5p.

***Autistic Disorders*:** miR-106b-5p, miR-128, miR-132-3p, miR-140-5p, miR-146b-5p, miR-148b-3p, miR-15a-5p, miR-15b-5p, miR-181d, miR-193b-3p, miR-212-3p, miR-27a-3p, miR-320a, miR-381-3p, miR-431-5p, miR-432-5p, miR-484, miR-539-5p, miR-652-3p, miR-7-5p, miR-93-5p, miR-95.

***Schizophrenia*:** let-7d-5p, let-7e-5p, miR-105-5p, miR-106b-5p, miR-107, miR-126-5p, miR-128, miR-130a-3p, miR-138-5p, miR-152, miR-15a-5p, miR-15b-5p, miR-181a-5p, miR-195-5p, miR-20a-5p, miR-212-3p, miR-24-3p, miR-26b-5p, miR-27a-3p, miR-29a-3p, miR-29b-3p, miR-29c-3p, miR-302a-5p, miR-302b-5p, miR-30d-5p, miR-338-3p, miR-346, miR-381-3p, miR-409-3p, miR-455-5p, miR-484, miR-485-5p, miR-487a, miR-489, miR-499a-5p, miR-512-3p, miR-518b, miR-7-5p, miR-9-3p, miR-92a-3p.

***Anxiety Disorder*:** miR-128, miR-485-3p, miR-509-3p.

***Tourette’s Syndrome*:** miR-24-3p.

***Prion Diseases*:** let-7b-5p, miR-128, miR-139-5p, miR-146a-5p, miR-191-5p, miR-203a, miR-320a, miR-337-3p, miR-338-3p, miR-339-5p, miR-342-3p.

***Huntington’s Disease*:** miR-124-3p, miR-132-3p, miR-135b-5p, miR-29a-3p, miR-29b-3p, miR-9-5p, miR-9-3p.

***Parkinson’s Disease*:** miR-133b, miR-433, miR-7-5p.

***Spinocerebellar Ataxia 1*:** miR-101-3p, miR-130a-3p, miR-19b-3p.

**Table 3 ijms-25-05826-t003:** Major miRNAs implicated in CNS development and neurogenesis.

miRNA	Target Molecule/Interaction	Effect	References
miR-124	SCP1	**Stimulation of neurogenesis, neuronal differentiation** During CNS development, timely down-regulation of SCP1 stimulates neurogenesis, and miR-124 contributes to this process by down-regulating SCP1 expression.	[152]
		**Nonneuronal cells including neural progenitors:** REST/SCP1 transcriptionally represses expression of miR-124 and other neuronal genes.	[153]
		**Neurogenesis:** miR-124 expression is derepressed, miR-124 post-transcriptionally suppresses multiple anti-neural factors including SCP1, resulting in further inhibition of the anti-neural pathway by REST/SCP1.	[154]
		miR-124 mediated repression of Sox9 associated with progression along the SVZ stem cell lineage to neurons (miR-124 is a neuronal fate determinant in the subventricular zone).	[155]
	Sox9	Neuronal differentiation as a result of interplay between miR-124, PTBP1, and SCP1/REST	[156]
			[157]
		De-repression of neuronal specific transcripts, including neurogenic RBPs *Nova1*, *Rbfox1* and the *nElavls*	[158]
		Stimulation of neurogenesis	
	*Ptbp1*		[159]
	*Zfp36l1*		[160]
	*Ezh2* (a negative regulator of neurogenesis)		[161]
microRNA-124-5p		Member of the synaptic microRNAome–regulators of the synaptic mRNA pool	[162]
miR-125a miR-125b	FMRP	Interaction with FMRP: regulation of the signal transduction of metabolic glutamate receptors (mGluR1) and N-methyl-D-aspartate receptors (NMDAR) and neuronal development	[163]
			[164]
miR-128 (critical role in cortical neurogenesis)	MSI1	Commitment of NSPC to the neuronal lineage	[165]
			[166]
	PCM1	Reduced NPC proliferation, stimulation of NPC differentiation into neurons	[167]
	Phf6	Cortical lamination: migration of neurons through the cortex, termination of upper neuron migration	[168]
miR-137	MSI1	Commitment of NSPC to the neuronal lineage	[165]
			[166]
		Neuronal differentiation and increased migration of progenitors into the cortical plate	[169]
	GluA1 subunit of AMPARs	Synaptic efficacy and mGluR-dependent synaptic plasticity	[170]
miR-9	*Zfp36* *Ezh2*	De-repression of neuronal specific transcripts, including neurogenic RBPs *Nova1*, *Rbfox1* and the *nElavls*; Stimulation of neurogenesis	[160]
miR-375	*Elavl4*	miR-375 is downregulated during the late stages of cortical development. The decrease of miR-375 leads to the de-repression of ELAVL4, with subsequent enhancement of neurite outgrowth in developing neurons	[171]
miR-132	p250GAP;	Regulation of dendritic growth and arborization of newborn neurons in the adult hippocampus (CREB-mediated signaling)	[172]
		A positive regulation of developing axon extension	[173]
	mRNA for the Ras GTPase activator Rasa1;	Synaptic structure and function	[174]
	FMRP		[175]
		Visual cortex plasticity	[164]
	MeCP2		
		Brain vascular integrity	[176]
	Cdh5		[177]
			[178]
			[179]
			[180]
			[181]
		Microglial homeostasis	[182]

**Table 4 ijms-25-05826-t004:** Neural differentiation- and cell proliferation-specific miRNAs and their targets.

miRNA	Targets	Some Important Roles
138	ARHGEF3, ROCK2, VIM, SIRT, ETC.	Precursor expressed in all tissues; mature miRNA only expressed in the brain. DNA Damage repair, and possibly sleep regulation.
26b	EPHA2, CDK6, CCNE1, ETC.	Neural Differentiation, and gene expression.
125b	IGF2, IL6R, E2F2, MAPK14, ETC.	Immune response, Osteoblast differentiation, and Neuroblastoma.
509	NTRK3, CFTR, ETC.	Cell proliferation and migration.
134	VEGFA, ABCC1, FOXM1, ETC.	Brain-specific, memory formation, and overexpressed in Schizophrenia.
132	SIRT, CDKN1A, CCNA2, ETC.	Neurogenesis, regulation of Inflammation in the brain and in the body, and Angiogenesis.
9	REST, NFKB1, SIRT1, VIM, ETC.	Neural Differentiation.
485	NTRK3, NFYB, ETC.	Synaptic formation regulation, and systemic iron balance.
128	RELN, TGFBR1, TP53, ETC.	Neuronal Migration, outgrowth, and excitability.
124	EFNB1, CDK4, CDK6, VIM, ROCK2, NR3C1, IT6B1, SLC16A1, ETC.	Neural Differentiation.

## Data Availability

This study collected demographic, behavioral, and laboratory data from normal, healthy women and from women who drank alcohol during pregnancy. Our research team supports all these activities and has developed a data-sharing plan. We also recognize that additional benefits from data sharing may arise in the future that are not apparent at this time, and we are prepared to work specifically with NIH in addressing all requests for raw data. At the present time, we have not deposited any of these raw data in an existing databank, but will make the data available to other investigators on request, in a manner consistent with NIH guidelines. Consistent with NIH policy, shared data will be rendered “free of identifiers that would permit linkages to individual research participants and variables that could lead to deductive disclosure of the identity of individual subjects” Intellectual property and data generated under this project will be administered in accordance with both University and NIH policies, including the NIH Data Sharing Policy and Implementation Guidance of 5 March 2003, and 0925-0001 and 0925-0002 (Rev 07/2022 through 01/31/2026). With this caveat observed, data will be made available to the NIH/NICHD/NIAAA. Sufficient identifiers will be provided to the NIH so that research participants can be assigned a Global Unique Identifier (GUID), which is a universal subject ID that protects personally identifiable information (PII). Using the GUID, NDAR can bring together multiple types of data collected from a single participant, regardless of where and when those data were collected. Biological samples (blood, serum, exosomes, and RNAs) and data that are shared will be completely free of identifiers that would permit linkages to individual research participants. We will make biological samples, deidentified data, and associated documentation available to users only under a data-sharing agreement that provides for (1) a commitment to using the data only for research purposes, (2) a commitment to securing the data using appropriate computer technology; and (3) a commitment to destroying or returning remaining samples after analyses are completed. Intellectual property and data generated under this project will be administered in accordance with both University and NIH policies, including the NIH Data Sharing Policy and Implementation Guidance of 5 March 2003. As the FAIR data bank receives approval from the NIH, the data will be made available to that group as well. The NIH will be implementing a new specific policy regarding data sharing https://grants.nih.gov/grants/guide/notice-files/NOT-OD-21-014.html, as of 25 January 2023. We will adopt that policy also. Data will be also available at https://www.mdpi.com/ethics accessed on 1 January 2025).

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
