# Peer review of "Fetal Brain-Derived Exosomal miRNAs from Maternal Blood: Potential Diagnostic Biomarkers for Fetal Alcohol Spectrum Disorders (FASDs)"

_ijms, 2024, doi:10.3390/ijms25115826_

Round 1
Reviewer 1 Report
Comments and Suggestions for Authors
Notes, questions and recommendations to the authors of the manuscript:
1. The topic of the manuscript is relevant, as the negative influence of alcohol on pregnancy and the fetus is increasing worldwide. In addition, fetal alcohol spectrum disorders (FASD), leading causes of neurodevelopmental impairment, cannot be diagnosed early in utero. The development contributes to the development of prenatal diagnosis of important syndromes and disorders. In this aspect, the research is within the scope of the scientific magazine International Journal of Molecular Sciences and more precisely in its Special Issue - Different Functions and Roles of microRNAs in Human Disease. That is why the article helps mankind to understand that alcoholism is not only a socially significant but also a health problem;
2. The abstract is not prepared according to the rules of the journal and exceeds the character limit. Please reorganize the summary! You could remove the section headings Introduction, Methods, Results, etc.!
3. Against the background of the impressive list of 224 authors, the Introduction is relatively modest as a section. Furthermore, the end of this section lacks a clearly stated research objective to match the title of the manuscript. Please complete the goal!
4. In the Results section, 4 tables, 1 scheme and 7 figures are used to present the experimental data. In my opinion, Scheme 1 does not belong in this section, as it provides valuable literature information on anatomical and molecular targets of neurological disease-associated miRNAs. You could move the scheme to the Introduction section. Also, you could increase the size by 10-15% in Figure 1 for better visualization;
5. Unlike the other sections, the Discussion is relatively fully developed and shows the professionalism and erudition of the authors. They are able to interpret their and the world's data on the problem in the area of the claim that microRNAs as a new indicator of alcohol-associated fetal pathology. Although the sample size was too small to control for all the potential variables that might affect fetal sensitivity to EtOH, such EtOH dose, maternal obesity, use of tobacco and other drugs of abuse, medications, and socioeconomic status, this is a basic study with many future prospects. I recommend that the authors use randomized trials, but also biochemical interpretation of the data, since microRNA expression has an important role in screening for socially relevant disorders and diseases, so that they can do follow-up studies to determine which children at risk actually develop FASD and possibly determine the molecular determinants of the subtypes and severity of FASD syndromes.
6. In the Material and methods section, the authors have complied with the requirements for the ethical norms for taking the biological material for research. In my opinion, the number of samples is too small, in addition, other factors such as age, smoking, amount of alcohol consumed, frequency of use, liver and kidney ethanol dehydrogenase activity, etc. were not taken into account. A larger randomized trial is needed to refine the details and circumstances to draw firmer conclusions;
7. The conclusions summarize what was achieved in the present study. The authors have endeavored to derive useful information and outline guidelines for future research to formulate more definitive useful recommendations for gynecological and pediatric practice.
Comments on the Quality of English LanguageNotes regarding the English language:
The article is written in relatively good and professional English. Final polishing of the manuscript by a professional English-speaking editor is required.
Author Response
Please see in the attached file

Reviewer 2 Report
Comments and Suggestions for Authors
I have reviewed the submission by Darbinian et al. It is overall difficult to follow the flow of information right from the abstract. Also, I find this study to contain superfluous citation of some very old literature-an otherwise problematic citing practice that distorts science, and which does not represent the current state of knowledge. I also note a sort of citation stuffing for the first author (Darbinian) and Armine Darbinyan (with 10 articles cited!). I encourage the authors to perform a very well thought of revision of the current draft, to polish it into a standard that is publishable in a journal with international readership.
SPECIFIC COMMENTS
1. Abstract (L15-L44): This needs to be summarized, by giving a brief background to the study, methodology applied, results, Implications and conclusion. In the current state, this is far from being called a report.
2. For Keywords, unusual abbreviations or words already captured in the title should preferably not be repeated as author-suggested indexing keywords as it reduces the discoverability of the final published article.
3. Introduction.
- Needs to be rethought about. The problem in question is UNIVERSAL, and should be introduced as such before talking about the US. It could already be pointed, for example, that the estimated global prevalence of FASD among the general population is 7.7 cases/1000 individuals. Actually, FASD prevalence is apparently highest in the WHO European Region (19.8 per 1,000) and lowest in the WHO Eastern Mediterranean Region (0.1 per 1,000). See Popova et al., 2023 (https://doi.org/10.1038/s41572-023-00420-x) for a better overview.
-L69, 73, 82-83, 444, 501: These lines contain unhealthy scientific citations, please revise them. Problematic citing practices always distort science! I also note a sort of citation stuffing for the first author Darbinian and Armine Darbinyan (with 10 articles cited), which is otherwise not acceptable as per Committee on Publication Ethics (COPE) guidelines. IJMS is a member of COPE (https://www.mdpi.com/journal/ijms/about), and has put in strict measures to reduce incidences of citation manipulation (citation stuffing/citation stacking, honorary citation, coercive citation manipulation, and citation cartels) as this compromise the integrity of research, may lead to journal delisting from important indexing databases or damage the reputation of the journal in the scientific community (https://doi.org/10.24318/cope.2019.3.1). I recommend that you revise the current draft keeping in mind the following points;
(a) Reduce as much as possible self-citations in the manuscript.
(b) Where this is unavoidable, ensure that the citation is genuine (i.e., the current manuscript is on a continuum of a long-term programme of research and previous publications are relevant for understanding the history of the accumulated record).
-L73-79 requires supporting citations.
-L79-80…. Between these lines, you need to highlight the fact that FASD cannot be diagnosed early in utero, giving the underlying challenges before talking about the role of MicroRNAs.
RESULTS
Figure 1 and 2 are very poorly visualized.
Figure 3 is not editable?
It is clearly hard to follow the results, making the discussion not feasible. A better approach would be to combine the RESULTS and DISCUSSION.
-The manuscript needs to be thoroughly checked by an academic who is proficient in English used in this field.
-Other suggestions are in the attached manuscript file.

Needs moderate revision
Author Response
Please see in the attached file

Reviewer 3 Report
Comments and Suggestions for Authors
Dear authors,
Thanks for the paper that is submitted that covers an interesting topic. I think this article attempts to review a new perspective on biomarkers in maternal blood for FASD. The article is well written and flows easily. I have major comments and edits that I would like to suggest to the authors:
After using an anti-plagiarism tool, I have found that there are literally copied some paragraphs and figures from the article published by some of the authors (Darbinian, Merabova) in December 2023:
Darbinian N, Merabova N, Tatevosian G, et al. Biomarkers of Affective Dysregulation Associated with In Utero Exposure to EtOH. Cells. 2023;13(1):2. Published 2023 Dec 19. doi:10.3390/cells13010002
The percentage of plagiarism for this article alone is 20%.
Also, there is another article published in December 2022 that covers the same topic, therefore, this article lacks novelty:
Darbinian N, Darbinyan A, Sinard J, et al. Molecular Markers in Maternal Blood Exosomes Allow Early Detection of Fetal Alcohol Spectrum Disorders. Int J Mol Sci. 2022;24(1):135. Published 2022 Dec 21. doi:10.3390/ijms24010135
Figure 6 and its findings are presented in this article and in the one submitted to review. The plagiarism rate for this article is 4%.
I attach the plagiarism report that shows this issue.
-The abstract should be shortened.
Self-citations:
I have noticed that most of the authors have huge amount of self-citations:
Darbinian 9
Darbinyam 6
Merubora 7
Tatevosian 5
Goetzl 6
Seltzer 6.
I think this circumstance is unethical and should not be performed.
Line 695: Reference should be modified
The conclusion section should not include any kind of reference.
I think the summary section should be removed.
Declaration of interest, institutional review board statement are duplicated

Author Response
Please see in the attached file

Round 2
Reviewer 2 Report
Comments and Suggestions for Authors
All my major concerns on the submission were answered. I find the current version suitable for publication
Comments on the Quality of English LanguageOnly minor fixes required
Author Response
Please see in the attached file

Reviewer 3 Report
Comments and Suggestions for Authors
Dear authors,
Thank you for taking the time to address comments on the manuscript. The manuscript has been greatly improved, but there are still some concerns about some issues:
The method section should be placed before the result section.
I have ethical concerns about the fact that these pregnant women drank alcohol during pregnancy. Knowing this issue, the reason is that there was no health advice. This circumstance is very important because these women could have had the opportunity to quit drinking with tailored health advice. Also, alcohol consumption was evaluated only using self-reported data, and not using biomarker analysis, so recall bias can be avoided. It is curious that a study that aims to introduce biomarkers to assess alcohol consumption in pregnancy does not use them as a comparison of these new ones.
What was the cause of elective termination of pregnancy? This circumstance could be a bias in the final analysis.
Author Response
Please see in the attached file

Round 3
Reviewer 3 Report
Comments and Suggestions for Authors
Dear authors,
Thanks again for addressing the new comments to the article.
The article has been greatly improved. Regarding the detection of alcohol consumption during pregnancy and as a suggestion, the detection of EtG in the hair sample can be useful in monitoring chronic consumption.
Author Response
Please see in the attached file
